# Converging Evidence Points to BDNF as Biomarker of Depressive Symptoms in Schizophrenia-Spectrum Disorders

**DOI:** 10.3390/brainsci12121666

**Published:** 2022-12-04

**Authors:** Mirko Manchia, Ulker Isayeva, Roberto Collu, Diego Primavera, Luca Deriu, Edoardo Caboni, Maria Novella Iaselli, Davide Sundas, Massimo Tusconi, Federica Pinna, Pasquale Paribello, Maria Scherma, Claudia Pisanu, Anna Meloni, Clement C. Zai, Donatella Congiu, Alessio Squassina, Walter Fratta, Paola Fadda, Bernardo Carpiniello

**Affiliations:** 1Unit of Psychiatry, Department of Medical Sciences and Public Health, University of Cagliari, 09124 Cagliari, Italy; 2Unit of Clinical Psychiatry, University Hospital Agency of Cagliari, 09124 Cagliari, Italy; 3Department of Pharmacology, Dalhousie University, Halifax, NS B3H 4R2, Canada; 4Division of Neuroscience and Clinical Pharmacology, Department of Biomedical Sciences, University of Cagliari, 09042 Cagliari, Italy; 5Tanenbaum Centre for Pharmacogenetics, Campbell Family Mental Health Research Institute, Centre for Addiction and Mental Health, Toronto, ON M6J 1H4, Canada; 6Department of Psychiatry, Institute of Medical Science, Laboratory Medicine and Pathobiology, University of Toronto, Toronto, ON M6J 1H4, Canada; 7Centre of Excellence “Neurobiology of Dependence”, University of Cagliari, 09124 Cagliari, Italy

**Keywords:** BDNF, schizoaffective disorder, schizophrenia, biomarker, genetics, depression

## Abstract

Brain-derived neurotrophic factor (BDNF) is a key modulator of neuroplasticity and has an important role in determining the susceptibility to severe psychiatric disorder with a significant neurodevelopmental component such as major psychoses. Indeed, a potential association between BDNF serum levels and schizophrenia (SCZ) and schizoaffective disorder (SAD) has been tested in diverse studies and a considerable amount of them found reduced BDNF levels in these disorders. Here, we aimed at testing the association of BDNF serum levels with several demographic, clinical, and psychometric measures in 105 patients with SCZ and SAD, assessing the moderating effect of genetic variants within the BDNF gene. We also verified whether peripheral BDNF levels differed between patients with SCZ and SAD. Our findings revealed that BDNF serum levels are significantly lower in patients affected by SCZ and SAD presenting more severe depressive symptomatology. This finding awaits replication in future independent studies and points to BDNF as a possible prognostic indicator in major psychoses.

## 1. Introduction

Brain-derived neurotrophic factor (BDNF) is a key modulator of neuroplasticity [1] and is expressed in almost all the cortical areas. A number of studies have shown alterations in BDNF levels in the peripheral and central nervous system (CNS), including the hippocampus and prefrontal cortex, which are brain regions critical for emotions, mood, and cognition [2]. BDNF is, with the nerve-growth factor, neurotrophin 3, and neurotrophin 4, one of the four neurotrophins identified in the mammalian brain [3]. BDNF seemingly plays an important role in different cellular processes, including neuronal differentiation, neurogenesis, and synaptic plasticity [4,5]. BDNF is initially synthesized as the precursor protein preproBDNF, and after undergoing cleavage, is secreted into the extracellular space as proBDNF and mature BDNF [6]. These two active forms elicit opposing biological effects, and while proBDNF form binds p75^NTR^ receptor, mature BDNF binds the high affinity tropomyosin-related kinase receptor type B (TrkB) [4,7]. As a result of its interaction with the p75^NTR^ receptor, the pro-BDNF promotes neuronal apoptosis [8], dendritic shrinkage, and hippocampal long-term depression (LTD) [9,10]. In contrast, mature BDNF-TrkB signaling promotes developmental processes of cell survival, proliferation, differentiation, and enhancement of synaptic plasticity, including long-term potentiation (LTP) [7,11]. BDNF is able to cross the blood-brain barrier [12], and peripheral BDNF levels are strongly associated with its concentrations in the CNS [13]. Importantly, BDNF modulates neural circuits that can, in turn, regulate dopaminergic, serotonergic and GABAergic systems [14,15,16,17], all neurochemical pathways putatively involved in the pathogenesis of psychiatric disorders. Therefore, several studies have addressed the possible role of BDNF in neuropsychiatric disorders, including schizophrenia (SCZ) spectrum disorders. The latter are severe neuropsychiatric disorders that are associated with gradual impairment in multiple domains of functioning. Their neurodevelopmental component appears prominent, and even though the first episode of psychosis is commonly observed in late adolescence or early adulthood, it is often preceded by prodromal symptoms that include nonspecific mood and behavioral changes, social withdrawal, and cognitive symptoms [18]. Neuroinflammation might be a key player of the pathophysiology of SCZ. Indeed, one mechanistic hypothesis involves the interplay of dysregulated cytokine levels and BDNF modulation [19]. The elevation of pro-inflammatory cytokines has been associated with reduced BDNF gene expression, which in turn might have an impact on cognitive decline in patients with SCZ [20]. A potential association between BDNF serum levels and SCZ has been tested in diverse studies, and a considerable amount of them found reduced BDNF levels in these patients [21,22,23,24,25,26]. Even though other studies did not observe any difference between BDNF serum levels of SCZ compared to healthy controls [27,28], results of two extensive meta-analytic studies showed a moderate decrease in serum and plasma BDNF levels of SCZ patients compared to healthy controls [29,30]. Further, another recent meta-analysis by Rodrigues-Amorim et al. [31] showed that there was a decline of peripheral BDNF levels in both drug-naïve and medicated SCZ patients. The findings also confirmed that BDNF levels were reduced throughout the disease course. Finally, the meta-analysis by Dombi et al. [32] confirmed that there is a moderate reduction in serum or plasma BDNF levels in patients with SCZ compared to healthy controls, less pronounced when cognition was considered. Overall, these results corroborate the role of BDNF as biomarker of illness activity in SCZ.

In this context, the primary aim of this study was to analyze the association of BDNF serum levels with several demographic, clinical, and treatment-related features, as well as with psychometric measures of cognitive function, premorbid developmental compromise, quality of life, subjective wellbeing, and side-effects. Further, we tested whether genetic variants within the BDNF gene would influence these associations. Our secondary aim was to test whether peripheral BDNF levels differed between patients with SCZ and SAD.

## 2. Materials and Methods

### 2.1. Design and Settings

The sample consisted of 105 patients with a diagnosis of SCZ or SAD confirmed using the Structured Clinical Interview for DSM-IV-TR Axis I Disorders, Patient Edition (SCID-I/P [33]. The patients were recruited at the community mental health center of the Unit of Psychiatry of the Department of Medical Science and Public Health, University of Cagliari and University of Cagliari Health Agency, Cagliari, Italy. They were between 18 and 64 years of age at the start of the study (1 September 2014), with or without other comorbid psychiatric disorders, and had to display stability of psychopathological symptoms for the previous six months. This cohort was constituted of all patients with diagnosis of schizophrenia or schizoaffective disorder who were followed up at the Unit of Psychiatry of the University of Cagliari and agreed to participate in the study. A proportion of patients did not consent to have their anthropometric measures (BMI, height) taken, resulting in missing data. Exclusion criteria were: (1) refusal to provide consent; (2) presence of acute psychopathological symptoms; (3) presence of illness-related cognitive impairment of such severity that affects their ability to cooperate; (4) presence of major unstable medical illness; (5) severe mental retardation; (6) major neurological disorder or previous head injury; (7) current drug and alcohol dependence, but not abuse. Written, informed consent was signed by all research participants. Participation was voluntary, and participants were allowed to withdraw at any point with no disadvantage to their treatments. All participants were examined by an expert clinical psychiatrist to certify that their capacity to consent was not compromised by their mental status. Twenty-five patients were treated with haloperidol, 24 with clozapine, 3 with amisulpride, 10 with aripiprazole, 1 with clotiapine, 3 with fluphenazine, 14 with olanzapine, 6 with paliperidone, 3 with periciazine, 4 with quetiapine, 9 with risperidone, 1 with zuclopentixole, while 2 were not treated with antipsychotics at the time of sampling. All the assessments were made by trained mental-health professionals (psychiatry residents and/or senior clinical staff). The complete and detailed protocol describing sociodemographic and clinical measures used in the study has been previously published [34].

### 2.2. Clinical Assessments

#### 2.2.1. Psychopathological Measures

Clinical symptomatology was assessed with the Positive and Negative Symptom Scale for Schizophrenia (PANSS) [35] and the Clinical Global Impression Scale for Schizophrenia (CGI-SCH) [36]. The PANSS is a 30-item scale containing three subscales measuring positive, negative, and general psychopathology symptoms. The CGI-SCH has been identified as a valid, reliable, and user-friendly alternative of other symptom severity measurement instruments, particularly in routine clinical practice [37]. The scale measures the severity of illness based on the scores of five subdomains, including the positive, negative, depressive, cognitive symptoms, and overall severity. The Premorbid Adjustment Scale (PAS) [38] consisting of childhood, early adolescence, late adolescence, adulthood, and general subscales, was used to understand if there was a relation between BDNF levels and premorbid adjustment, as premorbid dysfunction in patients with psychosis is well documented and such impairments might be endophenotypic markers of the disorder. The duration of untreated psychosis (DUP), that implies the period between the age of onset of full-blown psychotic symptoms and the first antipsychotic treatment, was assessed in a standardized way by using the Psychopathological Onset Latency and Treatment Questionnaire [39]. A longer DUP in patients with SCZ and SAD has been previously associated with higher risk of relapse, as well as worse short-term and long-term outcomes [40,41,42].

#### 2.2.2. Social Functioning and Quality of Life Measures

Evaluation of the social functioning was carried out with the Personal and Social Performance Scale (PSP) [43], and the quality of life was assessed with the WHO Quality of Life brief questionnaire (WHOQOL-Brief) [44]. PSP scale assesses social functioning based on four dimensions, which include socially useful activities, personal and social relationships, self-care, and disturbing and aggressive behaviors. The WHOQOL-Brief consists of four subdomains assessing physical health, psychological health, social relationships, and environmental health. The standardized assessment of subjective wellbeing was made with the Subjective Wellbeing under Neuroleptics-Short Version (SWN-S), consisting of 20 statements that assess five subdomains: mental functioning, self-control, emotional regulation, physical functioning, and social integration [45].

#### 2.2.3. Cognitive Assessment

Cognitive function was assessed with the Brief Assessment of Cognition in Schizophrenia Scale (BACS) [46]. The BACS has shown to be highly sensitive to the cognitive impairments in schizophrenia and includes the assessments of four neurocognitive domains including: reasoning and problem solving, processing speed, verbal memory, and working memory.

### 2.3. Assessment of BDNF Serum Levels and Genotyping

BDNF serum levels of the recruited patients were assessed using a commercial human enzyme-linked immunoassay (ELISA) kit (Booster Immunoleader, Cat. N° EK0307). This kit has a high sensitivity (<2 pg/mL), a measuring interval of 31.2–2000 pg/mL and is used for the quantitative detection of human BDNF in cell culture supernatants, serum, and plasma. Blood sample of each patient was drawn at the same time of the day (between 8:00 and 10:00 AM). After that, blood samples were kept in serum separator tubes at a room temperature for about 4 h to allow for clotting. Following that, these samples were centrifuged at approximately 1000× *g* for 15 min. Finally, the supernatant serum samples were stored in small aliquots at −20 °C for future analysis. Serum BNDF levels of the patients were determined by being processed as per the manufacturer’s protocol and kit instructions. Within 30 min after the final step of the kit procedure, the optical density absorbance of the samples was measured using a microplate reader (Thermo Scientific Multiskan FC, Waltham, MA, USA) set at 450 nm. Obtained data were analyzed using the Thermo Scientific SkanIt Software 3.0 for Multiskan FC.

We used the Tagger program implemented in the Haploview v4.2 to select SNPs in linkage disequilibrium (LD) (r^2^ ≥ 0.8), with a minor allele frequency threshold of 0.01. The genotyping of SNPs rs1519480, rs11030104, rs6265 (Val66Met), and rs7934165 was performed using TaqMan probes on demand (C_11592757_20, C_1751792_10, C_11592758_10, C_1197567_10, ThermoFisher Scientific) on a StepOne Plus instrument (ThermoFisher Scientific). The reaction mixture was prepared in a final volume of 10 μL, consisting of 5 μL of MasterMix (2×), 0.5 μL of Custom TaqMan^®^ SNP Genotyping Assay (20×) containing primers marked as VIC and FAM to discriminate between alleles, 1 μL of cDNA, and 3.5 μL of RNase-free water. Polymerase Chain Reaction (PCR) was performed with the following conditions: 30 s. 60 °C, 10 min 90 °C, 40 cycles at 95 °C for 15 s, and 60 °C for 1 min.

### 2.4. Statistical Analysis

We tested whether BDNF serum levels were associated with several demographic and clinical variables using univariate analysis. Specifically, we visually explored the relationship between BDNF serum levels and independent continuous variables using scatter plots. If a linear relationship was identified, we used Pearson’s product moment correlation coefficient or Spearman correlation (rho) to test their correlation, depending on their empirical distribution. In case of categorical independent variables, we used *t*-test or ANOVA to compare BDNF levels across groups. When data violated the assumption of normality, we applied nonparametric tests. To decrease the risk of Type I error, we used a more stringent *p* = 0.01 as a threshold for statistical significance. To avoid inflating the risk of Type II error, we reported results with 0.01 < *p* < 0.05 as trend associations. This approach remains conservative, avoiding over penalization by multiple testing correction. Variables associated with BDNF serum levels at either significance levels were then added into a multiple linear regression model to account for possible intercorrelations among independent variables.

We applied ANCOVA to test for the effect of genotypic variants on BDNF serum levels depending on the diagnosis. Quality control (Hardy Weinberg Equilibrium (HWE)) and the analysis of the genetic association of the four BDNF gene SNPs with SCZ/SAD and BDNF levels as a quantitative trait were performed with PLINK 2.0 [47]. Specifically, we used linear regression under additive genetic model. All remaining analyses were made with R (version 4.0.1) and IBM^®^ SPSS^®^ Statistics version 27.

## 3. Results

### 3.1. Sample Characteristics

The sample consisted of 105 patients, 64 with a diagnosis of SCZ and 41 with a diagnosis of SAD. The mean age of the sample was 48.85 ± 10.45 years. The demographic and clinical characteristics of the patients that participated in the study are detailed in Table 1.

### 3.2. Associations of Clinical Variables with BDNF Levels

The statistical properties of the psychometric measures applied in our sample are listed in Table 2.

We then tested for association between BDNF serum levels and sociodemographic, clinical, and psychometric measures using nonparametric Spearman correlation (rho) (Table 3). Univariate analysis showed significant correlations between BDNF serum levels and the CGI-SCH severity of depressive symptoms (rho = −0.32, *p* = 0.001). There was also significant association between the BACS, verbal fluency controlled oral word association test (rho = 0.23, *p* = 0.03), and BDNF serum levels. Specifically, we found that (1) the lower the levels of BDNF, the higher the severity of symptoms as expressed by CGI-SCH, and (2) the higher the levels of BDNF, the higher the performance in verbal fluency task of BACS. We also found a statistically significant correlation between BDNF serum levels and the score of PAS childhood subscale. No statistically significant association was found with BDNF serum levels when analyzing categorical variables. All these results are summarized in Table 3 and Table 4. Finally, to understand if BDNF serum levels of SCZ and SAD patients differ from each other, we performed t-test analysis, and the results showed that there was no statistically significant difference between them: t(79) = −0.88, *p* = 0.38.

### 3.3. Multiple Linear Regression Analysis

When we controlled for possible intercorrelations using a multiple linear regression model incorporating PAS childhood, BACS, verbal fluency (controlled oral word association test), and CGI-SCH severity of depressive symptoms, only the latter remained significantly associated with BDNF serum levels (β = −4.73, t = −2.4; *p* = 0.02). The variance explained by the model was not negligible with a R^2^ of 0.24.

### 3.4. Analysis of BDNF Genetic Variants

All SNPs were in HWE. The analysis of the association of the four SNPs with the diagnosis of SCZ and with BDNF serum levels considered as quantitative trait did not show statistically significant results under additive model (data not shown). ANCOVA models with BDNF serum levels as predictor outcomes accounting for the effect of diagnosis and genotype of each of the four SNPs also did not show statistically significant associations.

## 4. Discussion

In this study we sought to investigate the association between BDNF serum levels of patients with schizophrenia and schizoaffective disorder with various demographic, clinical, treatment-related, and genetic factors. Our results revealed that BDNF serum levels are significantly lower in patients affected by SCZ and SAD presenting more severe depressive symptomatology. The statistically significant correlations between BDNF serum levels and premorbid adjustment in childhood, as well as verbal fluency cognitive domain that were found using univariate analysis, were not confirmed when the variables were added to the multiple linear regression model. We also tested for a modulatory effect of genetic variants on BDNF serum levels accounting also for diagnosis, without identifying statistically significant differences.

The finding of decreased levels of BDNF in patients with more severe depressive symptomatology confirmed with the multiple linear regression (β = −4.73, t = −2.4) are concordant with the results of a recent study where the reduction of the BDNF levels was correlated to the dimensional function of the depression and was not related to positive, negative, or to other general symptoms of SCZ and SAD [48]. Lower BDNF levels have been associated with major depressive disorder (MDD) previously [13,49], and the results of recent meta-analysis showed that antidepressant treatment elevated BDNF levels in patients with MDD [50]. Another study that investigated BDNF levels of patients affected by SCZ in comparison to healthy subjects also found that SCZ patients with depressive symptoms displayed lower levels of plasma BDNF [51]. The precise mechanisms underlying the causality or the direction of the association between BDNF levels and depressive symptoms remain unclear, however, there are possible explanations. According to the neurotrophic hypothesis of depression [2,52], decreased neurotrophic support leads to neuronal atrophy and reduction in hippocampal neurogenesis, which could lead to the development of depression. Conversely, the results of another study showed that negative life events and depressive episodes contributed to a sharp decrease in BDNF levels, which might suggest that persistent depression in turn may also affect the reduction in BDNF levels [53].

In our study, the association of moderate magnitude (β = −4.73, t = −2.4) between depressive symptoms and lower BDNF serum levels was found when assessed by the CGI-SCH. Conversely, our analyses did not identify a statistically significant association between the PANSS scale (and sub items) and BDNF levels. The results of several previous studies also did not find correlations between peripheral BDNF levels and PANSS scores [24,54]. Another study reported that, after six weeks of antipsychotic treatment, PANSS scores of psychotic patients significantly decreased. However, no relationship between PANSS scores and peripheral BDNF levels was observed [55]. These results support evidence from a recent meta-analysis by Fernandes et al. [29], where no association between serum and plasma BDNF levels and PANSS scores was detected. Nonetheless, a few previous studies have found significant associations between PANSS scores and BDNF levels [21,56,57]. Significant negative correlations between serum BDNF levels and PANSS positive and negative subscale scores were found in a sample of drug-naïve first-episode patients with SCZ [56]. In contrast, increased BDNF levels in patients with higher positive PANSS scores was observed in another study [21]. The discrepancy in the results of the previous studies could be attributed to the clinical heterogeneity of the samples used in mentioned studies. There could be a possible explanation for CGI-SCH, but not PANSS, being able to detect association between depressive symptoms and BDNF serum levels in the current study. The PANSS scale has been previously criticized because of its complexity. Furthermore, the depression subscale of the PANSS scale was said to be problematic for not being able to distinguish between depression, negative symptoms, and extrapyramidal side effects [58]. In addition, factors such as medication status of the patient, illness duration, clinical subtypes of SCZ, media of BDNF measurement (serum or plasma), and others might affect the measurement of the relationship between severity of psychopathological symptoms and BDNF levels.

We also found a significant association between verbal fluency scores and BDNF serum levels when performed univariate analysis. Neurocognitive impairments have been previously associated with lower BDNF levels in schizophrenia spectrum disorders [59,60,61,62,63]. The results of a recent study showed that diminished cognitive function was associated with higher negative symptoms and low BDNF serum levels [64]. In one study, SCZ patients who engaged in ten weeks of computerized cognitive training demonstrated a significant increase in serum BDNF levels when compared to controls [61]. Furthermore, the findings of the recent longitudinal study of 12 weeks suggest that peripheral BDNF levels could be considered as potential biomarker for cognitive improvement in acute schizophrenia patients [62]. A recent cross-sectional study that used BACS as a measurement instrument to observe the relationship between cognitive impairment in SCZ and peripheral BDNF levels, found an association between them [59].

The secondary aim of our study was to examine if there were differences between BDNF levels of SCZ and SAD groups. We did not find any significant difference between BDNF serum levels of these patients. Even though SAD has been less studied than SCZ, and the nosology of SAD remains controversial, there is substantial evidence showing a partial genetic [65,66,67] and clinical [68,69] overlap between these disorders. In the future, it would be useful to do the subgroup analysis with a larger sample size, which is possibly more homogeneous clinically. It should be also noted that our sample was comprised of patients attending a community mental health center with varying illness duration, that in some cases lasted more than a decade (Table 1). Contrary to our expectations, our study did not identify a statistically significant association between a longer duration of illness and BDNF levels, a finding consistent with the recent literature [70]. Another finding related to antipsychotic treatment in our sample deserves to be discussed. The absence of an effect of treatment in our sample contrasts with the literature [29], although, similarly to our study, the effect of antipsychotics on BDNF was detectable in plasma but not in serum.

Our findings should be interpreted in the context of several limitations. The primary limitation of this study is the lack of healthy control group which does not allow one to infer whether the observed associations between BDNF serum levels and psychopathological measures are specific to the disorders. In addition, the limited sample size prevented additional subgroup analysis and might have impacted our ability to detect association signals of small to moderate magnitude. Conversely, the identified association is likely to be suitable for testing in additional samples given the relatively strong effect size. The limited sample size also impacted our ability to identify genetic association, which resulted negative. It is also possible that the cross-sectional design of the study can impact the reliability of the information collected (recall bias). On the other hand, it should be noted that the clinical data collected for these analyses are based on longitudinal follow ups that, in some instances, lasted decades. Finally, our study assessed only serum BDNF levels of patients with SCZ and SAD, as it has been determined that higher concentrations of BDNF are found in serum than in plasma [71,72]. There is a controversy about serum or plasma BDNF being used as more accurate proxy of BDNF levels in the CNS. Some studies argue that BDNF serum levels are more accurate measures of BDNF levels in the brain [73], while some suggest that they are less accurate measures of acute changes in the CNS in comparison to plasma BDNF levels [29]. However, there are several studies that claim that plasma and serum BDNF levels are two separate measures with diverse biological relevance [74,75]. One source of weakness in this study, which could have affected the measurements, was the usage of ELISA kits, which could not distinguish between proBDNF and mature BDNF. Mature BDNF-TrkB signaling is the process which regulates dendritic growth, cell survival, proliferation, differentiation, and enhancement of synaptic plasticity [76], while proBDNF has opposing effects [7]. Therefore, further research using ELISA assays that can differentiate between pro and mature BDNF is required to establish a greater degree of accuracy on this matter.

A final consideration concerning our findings should be made. Numerous studies have shown that BDNF is involved in several psychiatric disorders, including depression and SCZ [2,77,78,79]. There is a growing number of results showing that there are decreased peripheral levels of BDNF and lowered concentrations of BDNF in the CNS of SCZ. Its extensive role in neurobiological processes, such as synaptic regulation, neuroplasticity, and neurogenesis, suggests that BDNF might be a plausible candidate for a biomarker of disease activity in humans. It is, however, unclear whether these fluctuations in BDNF levels play a causal role in the development of SSDs or if they are a pathological epiphenomenon of the disease. Animal models show that mice with low BDNF levels show symptoms such as anxiety [80], memory impairment [81], weight gain [82], etc. The results of animal studies, as well as the results of the present study, seem to confirm that BDNF could play an important role in the development of affective and cognitive symptoms associated with neuropsychiatric disorders. This is supported by the previous evidence, where successful treatment resulted in partial normalization of BDNF levels [83,84].

## 5. Conclusions

In sum, our analysis found that BDNF serum levels were associated with more severe depressive symptomatology in patients affected by SCZ and SAD. This finding awaits replication in future independent studies and points to BDNF as a possible prognostic indicator in major psychoses.

## Figures and Tables

**Table 1 brainsci-12-01666-t001:** Demographic and clinical characteristics of 105 patients with schizophrenia and schizoaffective disorder.

**Variable (Continuous)**	**N**	**Mean**	**SD**
BDNF serum levels, ng/mL	105	25.45	13.67
Age, years	105	48.85	10.45
Education, years	105	9.26	3.23
Offspring, N	105	0.34	0.95
Age of onset, years	105	21.77	9.30
Duration of illness, months	105	308.51	134.33
Age at first treatment, years	105	24.23	8.95
Duration of untreated illness, months	105	29.07	54.60
Cigarettes smoked per day, N	46	19.83	9.80
Weight, Kg	74	77.32	18.76
Height, cm	74	167.84	8.83
BMI	73	27.30	6.80
Waist circumference, cm	45	90.47	18.37
Antipsychotics, chlorpromazine equivalents, mg/die	103	378.92	272.03
**Variable (categorical)**		**N**	**%**
Sex (male)		74	70.5
Marital status			
Single		8	7.6
Married/Cohabiting		10	9.5
Divorced		2	1.9
Widowed		83	79.0
NA		2	1.9
Presence of offspring		19	18.1
Employment			
Employed		7	6.7
Student		1	1.0
Registered disabled civilian		95	90.5
Unemployed		2	1.9
Presence of smoking		52	49.5
History of substance abuse		28	30.8
Current use of substances		5	5.5
Presence of family history of mental disorders		64	61.0
Presence of family history of schizophrenia		31	29.5
Presence of family history of bipolar disorder		8	7.6
Presence of family history of major depressive disorder		19	18.1
Presence of family history of anxiety disorders		10	9.5
Clinical course			
Episodic with full remission		2	1.9
Episodic with residual symptoms		37	35.2
Chronic with or without periodical relapses		65	61.9
NA		1	1.0
Presence of hospital admissions		93	88.6
Diagnosis of schizophrenia (SCID-I)		64	61.0
Diagnosis of schizoaffective disorder (SCID-I)		41	39.0
Diagnosis of obsessive-compulsive disorder (SCID-I)		5	4.8
Diagnosis of cluster A personality disorders (SCID-II)		2	1.9
Diagnosis of cluster B personality disorders (SCID-II)		2	1.9
Diagnosis of cluster C personality disorders (SCID-II)		2	1.9
Diagnosis of personality disorder NOS (SCID-II)		1	1.0
Long-acting antipsychotic therapy		24	22.9

BDNF: brain-derived neurotrophic factor; BMI: body mass index; SCID-I: Structured Clinical Interview for DSM-IV Axis I Disorders; SCID-II: Structured Clinical Interview for DSM-IV Axis II Disorders.

**Table 2 brainsci-12-01666-t002:** Statistical properties of the psychometric measures applied in the sample of 105 patients with schizophrenia and schizoaffective disorder.

Psychometric Measure	N	Mean	SD
PAS childhood	98	1.38	1.22
PAS early adolescence	98	2.05	1.30
PAS late adolescence	92	2.96	1.44
PAS adulthood	80	3.13	1.70
PAS general	97	3.43	1.24
PANSS, positive symptoms	105	14.84	4.76
PANSS, negative symptoms	105	19.38	6.56
PANSS, general psychopathology	105	40.70	10.41
PANSS, total score	105	74.92	17.93
CGI-SCH, severity positive symptoms	105	3.30	1.37
CGI-SCH, severity negative symptoms	105	3.25	0.94
CGI-SCH, severity depressive symptoms	105	2.36	1.05
CGI-SCH, severity cognitive symptoms	105	3.29	1.12
CGI-SCH, global severity	105	3.60	0.96
PSP, total score	105	50.60	14.18
WHOQOL, physical health	101	12.91	2.71
WHOQOL, psychological	101	12.08	1.78
WHOQOL, social relationship	101	11.68	3.74
WHOQOL, environment	101	12.11	2.35
BACS, verbal memory	102	5.26	2.34
BACS, digit sequencing task (number of correct response)	102	12.71	6.23
BACS, digit sequencing task (longest sequence recalled correctly)	102	5.08	2.09
BACS, verbal fluency (category instances)	102	9.14	5.05
BACS, verbal fluency (controlled oral word association test)	102	16.25	7.93
BACS, attention, and speed of information processing (symbol coding)	101	24.16	15.03
BACS, executive functions, Tower of London	101	8.71	6.40
SWN, mental functioning	100	15.97	4.23
SWN, self-control	100	15.94	3.24
SWN, physical functioning	100	16.29	4.33
SWN, emotional regulation	100	16.22	3.67
SWN, social integration	100	15.95	3.53
SWN, total score	100	80.37	13.96

PANSS: positive and negative syndrome scale; BMI: body mass index; PAS: premorbid adjustment scale; CGI-SCH: clinical global impression—schizophrenia; PSP: personal and social performance scale: WHOQOL: the World Health Organization quality of life scale; BACS: brief assessment of cognition in schizophrenia; SWN: subjective well-being under neuroleptics.

**Table 3 brainsci-12-01666-t003:** Association of BDNF serum levels with continuous demographic and clinical variables in 105 patients with schizophrenia and schizoaffective disorder.

Demographic and Clinical Variable (Continuous)	BDNF Serum Levels (ng/mL) (rho)	*p*	N
Age, years	−0.1	0.33	105
Education, years	−0.09	0.36	105
Offspring, N	−0.01	0.88	105
Age of onset, years	−0.07	0.48	105
Length of current episode, months	−0.01	0.9	105
Age at first treatment, years	−0.18	0.06	105
Duration of illness, months	−0.12	0.90	105
Duration of untreated illness, months	−0.07	0.49	105
Cigarettes smoked per day, N	−0.02	0.91	46
Weight, Kg	0.18	0.12	74
Height, cm	0.11	0.35	74
BMI	0.26	0.30	18
Waist circumference, cm	0.08	0.6	45
PAS childhood	**−0.28**	**0.01**	98
PAS early adolescence	−0.2	0.07	98
PAS late adolescence	−0.12	0.3	92
PAS adulthood	−0.12	0.4	80
PAS general	−0.07	0.53	97
PANSS, positive symptoms	0.03	0.75	105
PANSS, negative symptoms	−0.01	0.39	105
PANSS, general psychopathology	−0.004	0.97	105
PANSS, total score	−0.04	0.73	105
CGI-SCH, severity positive symptoms	0.08	0.43	105
CGI-SCH, severity negative symptoms	−0.19	0.06	105
CGI-SCH, severity depressive symptoms	**−0.32**	**0.001**	105
CGI-SCH, severity cognitive symptoms	−0.12	0.21	105
CGI-SCH, global severity	−0.16	0.09	105
PSP, total score	0.02	0.8	105
WHOQOL, physical health	−0.01	0.91	101
WHOQOL, psychological	0.1	0.31	101
WHOQOL, social relationship	0.07	0.47	101
WHOQOL, environment	0.13	0.2	101
BACS, verbal memory	−0.03	0.79	102
BACS, digit sequencing task (number of correct response)	0.06	0.54	102
BACS, digit sequencing task (longest sequence recalled correctly)	0.04	0.69	102
BACS, verbal fluency (category instances)	0.02	0.83	102
BACS, verbal fluency (controlled oral word association test)	*0.23*	*0.02*	102
BACS, attention, and speed of information processing (symbol coding)	0.005	0.96	101
BACS, executive functions, Tower of London	0.04	0.72	101
SWN, mental functioning	0.16	0.11	100
SWN, self-control	0.07	0.49	100
SWN, physical functioning	0.11	0.3	100
SWN, emotional regulation	0.10	0.34	100
SWN, social integration	0.11	0.29	100
SWN, total score	0.16	0.10	100
Antipsychotics, chlorpromazine equivalents, mg/die	0.08	0.38	103

BDNF: brain-derived neurotrophic factor; PANSS: positive and negative syndrome scale; BMI: body mass index; PAS: premorbid adjustment scale; CGI-SCH: clinical global impression—schizophrenia; PSP: personal and social performance scale: WHOQOL: the World Health Organization quality of life scale; BACS: brief assessment of cognition in schizophrenia; SWN: subjective well-being under neuroleptics. Significant *p* values are highlighted in bold. Trend *p* values are in italic.

**Table 4 brainsci-12-01666-t004:** Association of BDNF serum levels with categorical demographic and clinical variables in 105 patients with schizophrenia and schizoaffective disorder.

Clinical Variable (Categorical)		BDNF Serum Levels (ng/mL), Mean (SD)	t or F	*p*
Sex	Female	24.5 (13.3)	0.47	0.63
	Male	25.9 (13.9)		
Age class	18–20	32.0 (8.8)	1.6	0.5
	21–25	24.9 (12.3)		
	26–44	26.1 (14.7)		
	45–65	21.0 (14.0)		
Marital status	Single	29.5 (14.7)	0.9	0.6
	Married/Cohabiting	18.6 (11.4)		
	Divorced	30.5 (26.7)		
	Widowed	25.5 (13.6)		
Presence of offspring	Yes	25.2 (14.9)	0.73	0.9
	No	25.5 (13.5)		
Employment	Employed	28.5 (17.8)	0.25	0.9
	Student	25.8 (NA)		
	Registered disabled civilian	25.1 (13.6)		
	Unemployed	32.5 (4.6)		
Presence of family history of mental disorders	Yes	26.5 (14.1)	0.4	0.9
	No	24.1 (13.1)		
Presence of family history of schizophrenia	Yes	26.5 (14.8)	−0.5	0.6
	No	25.0 (13.3)		
Presence of family history of bipolar disorder	Yes	27.1 (9.5)	−0.5	0.6
	No	25.3 (14.0)		
Presence of family history of major depressive disorder	Yes	24.1 (14.2)	0.4	0.6
	No	25.7 (13.6)		
Presence of family history of anxiety disorders	Yes	26.0 (12.1)	−0.1	0.9
	No	25.4 (13.9)		
Clinical course	Episodic with full remission	32.6 (5.8)	0.1	0.9
	Episodic with residual symptoms	28.7 (14.7)		
	Chronic with or without periodical relapses	23.6 (12.9)		
Presence of hospital admissions	Yes	25.6 (13.9)	−0.3	0.8
	No	24.4 (11.9)		
Presence of smoking	Yes	27.1 (14.6)	−1.6	0.1
	No	22.4 (12.1)		
History of substance abuse	Yes	23.8 (13.1)	0.6	0.5
	No	25.7 (14.1		
Current use of substances	Yes	19.7 (10.9)	0.9	0.4
	No	25.5 (13.9)		
Diagnosis	Schizophrenia	24.9 (13.5)	0.5	0.6
	Schizoaffective disorder	26.3 (14.0)		
Diagnosis of cluster A personality disorders (SCID-II)	Yes	21.5 (14.2)	0.4	0.7
	No	25.6 (13.7)		
Diagnosis of cluster B personality disorders (SCID-II)	Yes	24.1 (15.4)	0.2	0.9
	No	25.6 (13.7)		
Diagnosis of cluster C personality disorders (SCID-II)	Yes	32.6 (23.6)	−0.43	0.5
	No	25.4 (13.6)		
Class of antipsychotics	First-generation	25.7 (13.01)	−0.15	0.88
	Second-generation	25.3 (13.9)		

BDNF: brain-derived neurotrophic factor; PANSS: positive and negative syndrome scale; BMI: body mass index; PAS: premorbid adjustment scale; CGI-SCH: clinical global impression—schizophrenia; PSP: personal and social performance scale: WHOQOL: the World Health Organization quality of life scale; BACS: brief assessment of cognition in schizophrenia; SWN: subjective well-being under neuroleptics.

## Data Availability

Not applicable.

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
