# Peer review of "Converging Evidence Points to BDNF as Biomarker of Depressive Symptoms in Schizophrenia-Spectrum Disorders"

_brainsci, 2022, doi:10.3390/brainsci12121666_

Round 1
Reviewer 1 Report
The authors have given insight into how neurotrophins are related to different symptom domains in schizophrenia spectrum disorder. The sample is adequately chosen and the methodology is clearly stated, so this process could be reproduced.
However, I would like to propose some suggestions for additional revision:
- The Introduction should also mention possible interactions of BDNF with cytokines in schizophrenia (for example, see Janicijevic S. et al., https://doi.org/10.1515/sjecr-2017-0031).
- In the Methodology, every single psychometric instrument used should be briefly explained (subdomains, scoring, etc.). Present the data about antipsychotic treatment.
- Explain in detail the multivariate linear regression model. The direction and strength of established correlations should be stated in the Results and mentioned in all future texts and Conclusions.
- The extensive part of the Discussion now seems to be commenting on the limitations of this study. The characteristics of the sample should be discussed, also the comparison of these subgroups of patients. Elaborate on the impact of diverse antipsychotics on BDNF. It should be considered the phase and duration of the illness. What mechanisms could underlie this association of BDNF with depressive symptoms but not the other domains.
- Explain the abbreviations in all table legends.
- Uniforming the references.
Author Response
Q1)The Introduction should also mention possible interactions of BDNF with cytokines in schizophrenia (for example, see Janicijevic S. et al., https://doi.org/10.1515/sjecr-2017-0031).
R1) We appreciate the reviewer’s suggestion and have added the following sentences on possible interactions of BDNF and cytokines in schizophrenia:
“Neuroinflammation might be a key player of the pathophysiology of SCZ. Indeed, one mechanistic hypothesis involves the interplay of dysregulated cytokine levels and BDNF modulation [19]. The elevation of pro-inflammatory cytokines has been associated with reduced BDNF gene expression, which in turn might have an impact on cognitive decline in patients with SCZ [20].”
Q2) In the Methodology, every single psychometric instrument used should be briefly explained (subdomains, scoring, etc.). Present the data about antipsychotic treatment.
R2) Thank you for pointing this out. As suggested by the reviewer we added a brief description of each psychometric instrument in the methodology section. Data on antipsychotic treatment were reported already in the Table 1 and Table 3. We added the division of the patients based on the main antipsychotic treatment: “Twenty-five patients were treated with haloperidol, 24 with clozapine, 3 with amisulpride, 10 with aripiprazole, 1 with clotiapine, 3 with fluphenazine, 14 with olanzapine, 6 with paliperidone, 3 with periciazine, 4 with quetiapine, 9 with risperidone, 1 with zuclopentixole, while 2 were not treated with antipsychotics at the time of sampling.”
Q3) Explain in detail the multivariate linear regression model. The direction and strength of established correlations should be stated in the Results and mentioned in all future texts and Conclusions.
R3) More details were added in the text as requested: “When we controlled for possible intercorrelations using a multiple linear regression model incorporating PAS childhood, BACS, verbal fluency (controlled oral word association test) CGI-SCH severity of depressive symptoms, only the latter remained significantly associated with BDNF serum levels (β = -4.73, t = -2.4; p = 0.02). The variance explained by the model was not negligible with an R2 of 0.24.” The remaining text was revised accordingly. As an aside we changed the terminology using the more appropriate term “multiple regression” instead of “multivariate” throughout the text.
Q4) The extensive part of the Discussion now seems to be commenting on the limitations of this study. The characteristics of the sample should be discussed, also the comparison of these subgroups of patients. Elaborate on the impact of diverse antipsychotics on BDNF. It should be considered the phase and duration of the illness. What mechanisms could underlie this association of BDNF with depressive symptoms but not the other domains.
R4) These are excellent observations, and we added the following statements in the Discussion:
For discussion of clinical variables: “In the future it would be useful to do the subgroup analysis with larger sample size, possibly more homogeneous clinically. It should be also noted that our sample was comprised of patients attending a community mental health center with varying illness duration, that in some cases lasted more than a decade (Table 1). Contrary to our expectations, our study did not identify a statistically significant association between a longer duration of illness and BDNF levels, a finding consistent with recent literature [51]. Another finding related to antipsychotic treatment in our sample deserves to be dis-cussed. The absence of an effect of treatment in our sample contrasts with the literature [29], although, similarly to our study, the effect of antipsychotics on BDNF was detectable in plasma but not in serum.”
For mechanisms underlying the association of BDNF with depressive symptoms please see: “The precise mechanisms underlying the causality or the direction of the association between BDNF levels and depressive symptoms remain unclear, however, there are possible explanations. According to the neurotrophic hypothesis of depression [2,38], decreased neurotrophic support leads to neuronal atrophy and reduction in hippocampal neurogenesis which could lead to the development of depression. Conversely, the results of another study showed that negative life events and depressive episodes contributed to a sharp decrease in BDNF levels, which might suggest that persistent depression in turn may also affect the reduction in BDNF levels [39].”
Q5) Explain the abbreviations in all table legends.
R5) Thank you for this observation. We apologize for this oversight. We have revised the abbreviations and added the missing ones.
Q6) Uniforming the references.
R6) Thank you for pointing this out. We have updated the references.
Reviewer 2 Report
The manuscript of Mirko Manchia et al. entitled „Converging evidence points to BDNF as biomarker of depressive symptoms in schizophrenia-spectrum disorders” reveals a possible new laboratory indicator of depression in schizophrenia and schizoaffective disorder. The manuscript is generally well-written and the interpretation of data (including limitations) is correct. I have some minor suggestions to improve the manuscript.
1. In materials and methods, it shold be indicated, whether the 105 patients were the original count of subjects, or more patients have been recruited, but the others were eliminated from the study (e.g. withdrawal, mived to other city, death, etc.). Similarly, some data (body weight, height, BMI etc. are available only about 73-74 patients, why? No consent or other reason?
2. It should be clarified, that drug or alcohol dependence were exlusion criteria, but substance use was not (as written in table 1).
3. Style of abbreviations should be uniform, e.g. preproBDNF, proBDNF vs. pro-BDNF in line 45-46. Similarly, in the list of references [1] and [2] seem to be in different style, than the other ones. Line 174, 175: the „(Hardy Weinberg Equilibrium (HWE)]” should be either (…) or […].
Author Response
Q1) In materials and methods, it should be indicated, whether the 105 patients were the original count of subjects, or more patients have been recruited, but the others were eliminated from the study (e.g. withdrawal, mived to other city, death, etc.). Similarly, some data (body weight, height, BMI etc. are available only about 73-74 patients, why? No consent or other reason?
R1) This cohort was constituted of all patients with diagnosis of schizophrenia or schizoaffective disorder who were followed up at the Unit of Psychiatry of the University of Cagliari and agreed to participate in the study. Data are missing for those patients who did not consent to have anthropometric measures taken. This has been clarified as follows: “This cohort was constituted of all patients with diagnosis of schizophrenia or schizoaffective disorder who were followed up at the Unit of Psychiatry of the University of Cagliari and agreed to participate in the study. A proportion of patients did not consent to have their anthropometric measures (BMI, height) taken resulting in missing data.”
Q2) It should be clarified, that drug or alcohol dependence were exclusion criteria, but substance use was not (as written in table 1).
R2) This has been clarified in the revised version of the manuscript: “7) current drug and alcohol dependence, but not abuse.”
Q3) Style of abbreviations should be uniform, e.g. preproBDNF, proBDNF vs. pro-BDNF in line 45-46. Similarly, in the list of references [1] and [2] seem to be in different style, than the other ones. Line 174, 175: the „(Hardy Weinberg Equilibrium (HWE)]” should be either (…) or […].
R3) We thank the reviewer for this observation. We have revised the text and all the above-mentioned items have been uniformed.